# EGFR testing and erlotinib use in non-small cell lung cancer patients in Kentucky

**Kara L. Larson**[1], **Bin Huang**[1,2,3], **Quan Chen**[1], **Thomas Tucker**[1,4], **Marissa Schuh**[1], **Susanne M. Arnold**[1,5], **Jill M. Kolesar**[1,5]*

1 Markey Cancer Center, University of Kentucky, Lexington, Kentucky, United States of America, 2 Division of Cancer Biostatistics, University of Kentucky, Lexington, Kentucky, United States of America, 3 Department of Internal Medicine, University of Kentucky, Lexington, Kentucky, United States of America, 4 Department of Epidemiology, University of Kentucky, Lexington, Kentucky, United States of America, 5 Department of Pharmacy Practice and Science, University of Kentucky, Lexington, Kentucky, United States of America

* jill.kolesar@uky.edu

**Data Availability Statement:** Data cannot be publicaly shared because they are both potentially identifying and contain sensitive patient data, including geographic location, dates of diagnosis and dates of testing and receiving a medication. In

## Abstract

This study determined the frequency and factors associated with EGFR testing rates and erlotinib treatment as well as associated survival outcomes in patients with non small cell lung cancer in Kentucky. Data from the Kentucky Cancer Registry (KCR) linked with health claims from Medicaid, Medicare and private insurance groups were evaluated. EGFR testing and erlotinib prescribing were identified using ICD-9 procedure codes and national drug codes in claims, respectively. Logistic regression analysis was performed to determine factors associated with EGFR testing and erlotinib prescribing. Cox-regression analysis was performed to determine factors associated with survival. EGFR mutation testing rates rose from 0.1% to 10.6% over the evaluated period while erlotinib use ranged from 3.4% to 5.4%. Factors associated with no EGFR testing were older age, male gender, enrollment in Medicaid or Medicare, smoking, and geographic region. Factors associated with not receiving erlotinib included older age, male gender, enrollment in Medicare or Medicaid, and living in moderate to high poverty. Survival analysis demonstrated EGFR testing or erlotinib use was associated with a higher likelihood of survival. EGFR testing and erlotinib prescribing were slow to be implemented in our predominantly rural state. While population-level factors likely contributed, patient factors, including geographic location (areas with high poverty rates and rural regions) and insurance type, were associated with lack of use, highlighting rural disparities in the implementation of cancer precision medicine.

## Introduction

Lung cancer is the leading cause of cancer death in the United State [1], and Kentucky leads the nation in both the rate of new cases and deaths due to cancer, with the Appalachian region carrying the highest cancer burden [2–4]. The high incidence and death rates in Kentucky demonstrate a clear need for more effective interventions in lung cancer patients.

Clinical studies associating EGFR mutations with better response to tyrosine kinase inhibitors were reported in 2004 [5–7]. Ongoing clinical trials at that time did not require the

addition, there are contractual agreements between the University of Kentucky and the Kentucky Cancer Registry precluding data sharing. Any requests for data must be submitted to: Jacyln K. McDowell, Epidemiologist, Kentucky Cancer Registry 2365 Harrodsburg Rd, Suite A230 Lexington, KY 40504 859-218-2228

**Funding:** JMK P30 CA177558 National Cancer Institute cancer.org The funders had no role in study design, data collection and analysis, decision to publish, or preparation of the manuscript.

**Competing interests:** The authors have declared that no competing interests exist.

presence of an EGFR mutation as an inclusion criteria, and erlotinib was initially approved in late 2004 as a monotherapy for the treatment of patients with locally advanced or metastatic non-small cell lung cancer (NSCLC) after failure of at least one prior chemotherapy regimen. The approval was based on the BR-21 trial, which compared erlotinib to placebo and demonstrated survival was significantly longer for patients treated with erlotinib. Multivariate analyses showed improved survival with erlotinib in the EGFR-positive group by immunohistochemistry, however since the multivariate analyses failed to rule out a small erlotinib survival effect in patients who were EGFR-negative, erlotinib was approved regardless of EGFR status [8].

The first EGFR mutation test was commercialized in 2005, however EGFR testing recommendations were not included in the American Society of Clinical Oncology (ASCO) and National Comprehensive Cancer Network (NCCN) guidelines until 2010 [9, 10]. The OPTIMAL and EUROTAC trials, which compared erlotinib to standard doublet chemotherapy in patients with EGFR mutations, demonstrated both improved progression-free survival and reduced adverse effects in the erlotinib arms. These were published in 2011 and 2012 and supported a new erlotinib indication in the front-line setting for EGFR mutant locally advanced or metastatic NSCLC [11, 12]. Erlotinib indications were updated again in 2016 with the publication of the IUNO trial, which demonstrated no survival benefit in EGFR wild-type individuals, and currently erlotinib is only approved in NSCLC for patients with an EGFR mutation [13]. Current guidelines published by ASCO recommend that all patients with advanced non-squamous NSCLC, regardless of clinical characteristics such as age, race, or smoking status should undergo testing for EGFR and other actionable mutations. [14].

Despite the availability of an EGFR mutation test as early as 2005 and recommendations for routine EGFR mutation analysis as a part of standard care, not all patients are tested. A 2010 NCCN survey found that less than 50% of oncologists tested their patients for EGFR mutations, and that less than 50% of patients who received erlotinib had EGFR testing done. The same study found that age, location, comorbidity scores, and treatment history of radiation therapy affected whether or not patients received the testing [15]. A later survey found that lack of test availability, unfamiliarity with testing benefits, inadequate tissue for testing, patient refusal, or a lack of access to targeted clinical trials resulted in low mutation testing rates [16].

The purpose of this study is to evaluate EGFR testing and erlotinib use in patients with NSCLC in Kentucky and identify factors associated with lack of testing or erlotinib treatment and associated survival.

## Materials and methods

### Setting

The Kentucky Cancer Registry (KCR) is a population-based central cancer registry for the Commonwealth of Kentucky. All healthcare facilities that diagnose or treat cancer patients, including all acute care hospitals and associated outpatient facilities, freestanding treatment centers, private pathology laboratories, and physician offices, are required to report each case of cancer to the KCR. The KCR has been part of the Centers for Disease Control and Prevention (CDC) National Program of Cancer Registries since 1994 and the National Cancer Institute's (NCI) Surveillance and Epidemiology and End Results (SEER) program since 2000. KCR has received the highest level of certification from the North American Association of Central Cancer Registries (NAACCR) indicating its commitment to accuracy, completeness, and quality [17].

KCR performed a probabilistic data linkage to identify matches between KCR and claims from Medicaid, state employee insurance and private insurance groups for cancer cases

diagnosed in 2000–2012. Medicare claims were also acquired from the SEER Medicare database. The final data set consolidated the linked claims data, including cancer cases diagnosed in 2000–2011, and claims up to 2015 from sources mentioned above [18].

## Study population

The cohort was selected from KCR with claims for cases diagnosed in 2007–2011. Patients must have presented with invasive NSCLC (Stage IIIB–Stage IV), have had continuous healthcare coverage one month prior to the date of diagnosis and one year after, and must have linked claims data. Over this time period, 5.3% of diagnosed cases occurred in uninsured individuals who were excluded from the analysis. Genetic test claims were captured within one month prior to diagnosis and three months after. Drug claims were captured within one year of diagnosis and could have been any line treatment (Table 1). The final cohort included 4957 individuals.

Demographics variables were extracted from the linked KCR data, including age at diagnosis, race, sex, smoking status, education, poverty status, metropolitan status, Appalachian status, insurance type, comorbidity, hospital type and distance to a hospital. Education level and poverty status were determined by percentage of high school completion rate and percentage of population below poverty range based on the 2000 US Census county estimates, then categorized into four levels based on the quartiles of their corresponding distributions. Metropolitan status was defined based on the 2013 Rural-Urban County Continuum Codes with values 1–3 as Metro and 4–9 as Non-Metro (https://www.ers.usda.gov/data-products/rural-urban-continuum-codes.aspx). Appalachian status was determined by the Appalachia Regional Commission (https://www.arc.gov/appalachian_region/CountiesinAppalachia.asp). A variable with the combination of metro and Appalachian status was also created. The reporting hospitals were categorized into two types: tertiary academic hospital (University of Kentucky and University of Louisville) or not. Carlson comorbidity index was calculated from the linked claims data. Using a Great Circle Distance approach, distance between patient residence and their corresponding hospital was calculated based the geocodes of the locations. Current Procedural Terminology (CPT) codes and National Drug Codes (NDC) were extracted from claims to identify the EGFR mutation test and erlotinib prescription.

## Statistical analysis

Descriptive analysis of the demographic and clinical factors was performed. $\chi^2$ tests were used to examine associations between demographic/clinical factors and EGFR test/erlotinib prescription. Two logistic regressions were fitted separately to identify significant factors associated with EGFR test or erlotinib prescription while controlling for other covariates. Kaplan-Meier plots and Cox regression survival analysis were also performed to examine how EGFR testing and erlotinib affect overall survival. The final models only kept variables with p-value < 0.1. All analyses were done using SAS Statistical software version 9.4 (SAS Institute,

**Table 1. Codes used to identify EGFR testing and erlotinib.**

|  | Code Type | Codes Used |
|---|---|---|
| Erlotinib | NDC | 69189–0063, 50242–062, 50242–063, 50242–064, 54868–5290, 54868–5447, 54868–5474, 54569–5848, 54569–5847 |
| EGFR | CPT | 81235,83891, 83894, 83896, 83898, 83903, 83904, 83907, 83909, 83912, 83890, 81401,83969 |

Inc., Cary, North Carolina, USA). All statistical tests were two sided with a P-value $\leq 0.05$ used to identify statistical significance.

### Ethical considerations

This study was approved by the University of Kentucky IRB #51483. Informed consent was waived as all data was de-identified before analysis All data was treated highly as confidential and was only accessible in password-protected files for authorized study staff.

### Results

From 2007 to 2011 the percentage of patients presenting with locally advanced or advanced stage disease that were tested for EGFR mutations increased from 0.1% to 10.6% (Table 2), while erlotinib use ranged from 3.4% to 5.4% with no trend over time. Demographics, including younger age, female gender, non-smokers and not being white or black were associated with EGFR testing and erlotinib prescribing. Individuals living in areas with high poverty, low high school attainment, and with Medicare or Medicaid insurance were significantly less likely to have EGFR testing or an erlotinib prescription. Geographic factors, both distance to an academic medical center and rural Appalachia, were significantly associated with EGFR testing, but not erlotinib prescribing.

Factors associated with EGFR testing were assessed through multi-variate logistic regression analysis (Table 3). Clinical variables, including age, gender and smoking status were associated with EGFR testing with younger, female, non-smokers more likely to be tested. Additionally, with the exception of 2008, the testing likelihood increased significantly for each year, 2009 (OR = 22.30, CI = 3.00 to 165.41), 2010 (OR = 58.56, CI = 8.12 to 422.26), and 2011 (OR = 113.47, CI = 15.81 to 814.21) compared to 2007 (P = <0.0001) despite overall rates remaining low. The variables measuring disparities were also significantly associated with a decreased likelihood of receiving testing. Patients enrolled in Medicaid (OR = 0.19, CI = 0.09 to 0.40) or Medicare (OR = 0.61, CI = 0.44 to 0.84) compared to those with private insurance (P = <0.0001) were less likely to receive testing. Those patients living in non-metropolitan areas, whether in Appalachian (OR = 0.51, CI = 0.36 to 0.73) or non-Appalachian regions (OR = 0.60, CI = 0.40 to 0.89), were also significantly less likely to receive testing (P = 0.0011).

To determine factors associated with erlotinib prescribing, the same variables were examined through multivariate logistic regression analysis (Table 4). Similarly, younger patients and female patients were more likely to receive the drug. In addition, those patients enrolled in Medicaid (OR = 0.55, CI = 0.33 to 0.93) and Medicare (OR = 0.63, CI = 0.46 to 0.87) were significantly less likely to receive the drug compared to those enrolled in private insurance (P = 0.0074). Those patients living in areas with moderate (OR = 1.90, CI = 1.24 to 2.91) and high poverty (OR = 1.84, CI = 1.22 to 2.79) were also significantly less likely to receive the drug compared to those living in low poverty (P = 0.0081).

Cox-regression survival analysis was performed to determine factors associated with likelihood of survival in patients with Stage IIIb–IV NSCLC (Table 5). The clinical characteristics associated with improved survival include younger age, female gender and a low co-morbidity score. Several other variables predicted survival. When comparing patients living in metropolitan Appalachia (HR = 1.09, CI = 0.93 to 1.28), rural Appalachia (HR = 1.10, CI = 0.97 to 1.25), and rural non-Appalachian Kentucky (HR = 1.13, CI = 1.04 to 1.23), patients living in rural, non-Appalachian regions had a significantly decreased likelihood of survival compared to those living in a metropolitan region (P = 0.0372). Furthermore, patients enrolled in Medicaid (HR = 1.17, CI = 1.05 to 1.31) and Medicare (HR = 1.11, CI = 1.03 to 1.19) had a significantly lower likelihood survival of compared to those with private insurance survival (P = 0.0053).

**Table 2. Bivariate analysis for EGFR testing and erlotinib receipt among NSCLC Stage III and IV patients.**

| | Had EGFR Testing | | | | | Received Erlotinib | | | | |
|---|---|---|---|---|---|---|---|---|---|---|
| | **No** | **%** | **Yes** | **%** | **P** | **No** | **%** | **Yes** | **%** | **P** |
| **Total** | **4748** | **95.8%** | **209** | **4.2%** | | **4744** | **95.7%** | **213** | **4.3%** | |
| **Age** | | | | | 0.0072 | | | | | 0.0058 |
| 20–49 | 162 | 91.0% | 16 | 9.0% | | 167 | 93.8% | 11 | 6.2% | |
| 50–64 | 999 | 95.4% | 48 | 4.6% | | 988 | 94.4% | 59 | 5.6% | |
| 65–74 | 1976 | 95.9% | 85 | 4.1% | | 1969 | 95.5% | 92 | 4.5% | |
| 75+ | 1611 | 96.4% | 60 | 3.6% | | 1620 | 96.9% | 51 | 3.1% | |
| **Gender** | | | | | <0.0001 | | | | | 0.0046 |
| Male | 2811 | 96.7% | 96 | 3.3% | | 2802 | 96.4% | 105 | 3.6% | |
| Female | 1937 | 94.5% | 113 | 5.5% | | 1942 | 94.7% | 108 | 5.3% | |
| **Race** | | | | | 0.0924 | | | | | <0.0001 |
| White | 4438 | 95.7% | 197 | 4.3% | | 4441 | 95.8% | 194 | 4.2% | |
| Black | 299 | 96.8% | 10 | 3.2% | | 294 | 95.1% | 15 | 4.9% | |
| Other | 11 | 84.6% | 2 | 15.4% | | 9 | 69.2% | 4 | 30.8% | |
| **Stage** | | | | | 0.1729 | | | | | 0.2765 |
| Stage IIIb and effusion | 278 | 94.2% | 17 | 5.8% | | 286 | 96.9% | 9 | 3.1% | |
| Stage IV | 4470 | 95.9% | 192 | 4.1% | | 4458 | 95.6% | 204 | 4.4% | |
| **Metro Status** | | | | | 0.0001 | | | | | 0.5738 |
| Metro | 2291 | 94.7% | 129 | 5.3% | | 2312 | 95.5% | 108 | 4.5% | |
| Non-Metro | 2457 | 96.8% | 80 | 3.2% | | 2432 | 95.9% | 105 | 4.1% | |
| **Appalachia Status** | | | | | 0.0053 | | | | | 0.1029 |
| Appalachia | 1624 | 96.9% | 52 | 3.1% | | 1615 | 96.4% | 61 | 3.6% | |
| Non-Appalachia | 3124 | 95.2% | 157 | 4.8% | | 3129 | 95.4% | 152 | 4.6% | |
| **Appalachia and Metro Status** | | | | | 0.0010 | | | | | 0.0629 |
| Appalachia Metro | 166 | 96.5% | 6 | 3.5% | | 171 | 99.4% | 1 | 0.6% | |
| Appalachia Non-Metro | 1458 | 96.9% | 46 | 3.1% | | 1444 | 96.0% | 60 | 4.0% | |
| Non-Appalachia Metro | 2125 | 94.5% | 123 | 5.5% | | 2141 | 95.2% | 107 | 4.8% | |
| Non-Appalachia Non-Metro | 999 | 96.7% | 34 | 3.3% | | 988 | 95.6% | 45 | 4.4% | |
| **Year of Diagnosis** | | | | | <0.0001 | | | | | 0.2454 |
| 2007 | 858 | 99.9% | 1 | 0.1% | | 823 | 95.8% | 36 | 4.2% | |
| 2008 | 944 | 99.6% | 4 | 0.4% | | 910 | 96.0% | 38 | 4.0% | |
| 2009 | 914 | 97.5% | 23 | 2.5% | | 886 | 94.6% | 51 | 5.4% | |
| 2010 | 1092 | 94.1% | 69 | 5.9% | | 1121 | 96.6% | 40 | 3.4% | |
| 2011 | 940 | 89.4% | 112 | 10.6% | | 1004 | 95.4% | 48 | 4.6% | |
| **Insurance Type** | | | | | <0.0001 | | | | | <0.0001 |
| Private | 966 | 93.0% | 73 | 7.0% | | 972 | 93.6% | 67 | 6.4% | |
| Medicaid | 502 | 98.2% | 9 | 1.8% | | 490 | 95.9% | 21 | 4.1% | |
| Medicare | 3280 | 96.3% | 127 | 3.7% | | 3282 | 96.3% | 125 | 3.7% | |
| **High School Attainment** | | | | | <0.0001 | | | | | 0.0176 |
| Very Low | 1191 | 96.4% | 44 | 3.6% | | 1193 | 96.6% | 42 | 3.4% | |
| Low | 1203 | 97.3% | 33 | 2.7% | | 1174 | 95.0% | 62 | 5.0% | |
| Moderate | 1171 | 96.0% | 49 | 4.0% | | 1179 | 96.6% | 41 | 3.4% | |
| High | 1183 | 93.4% | 83 | 6.6% | | 1198 | 94.6% | 68 | 5.4% | |
| **Poverty** | | | | | 0.0122 | | | | | 0.0032 |
| Low | 1179 | 96.0% | 49 | 4.0% | | 1193 | 97.1% | 35 | 2.9% | |
| Moderate | 1062 | 94.1% | 66 | 5.9% | | 1066 | 94.5% | 62 | 5.5% | |
| High | 1301 | 96.0% | 54 | 4.0% | | 1285 | 94.8% | 70 | 5.2% | |

*(Continued)*

**Table 2.** (Continued)

| | Had EGFR Testing | | | | | Received Erlotinib | | | | |
|---|---|---|---|---|---|---|---|---|---|---|
| | **No** | **%** | **Yes** | **%** | **P** | **No** | **%** | **Yes** | **%** | **P** |
| Very High | 1206 | 96.8% | 40 | 3.2% | | 1200 | 96.3% | 46 | 3.7% | |
| **Charlson Comorbidity Index** | | | | | 0.3214 | | | | | 0.1085 |
| 0 | 2074 | 95.2% | 104 | 4.8% | | 2071 | 95.1% | 107 | 4.9% | |
| 1 | 1328 | 96.0% | 56 | 4.0% | | 1327 | 95.9% | 57 | 4.1% | |
| 2 | 682 | 96.5% | 25 | 3.5% | | 677 | 95.8% | 30 | 4.2% | |
| 3+ | 664 | 96.5% | 24 | 3.5% | | 669 | 97.2% | 19 | 2.8% | |
| **Smoking** | | | | | 0.0393 | | | | | 0.0088 |
| No | 258 | 92.8% | 20 | 7.2% | | 256 | 92.1% | 22 | 7.9% | |
| Yes | 4052 | 96.0% | 170 | 4.0% | | 4051 | 95.9% | 171 | 4.1% | |
| Unknown | 433 | 95.8% | 19 | 4.2% | | 432 | 95.6% | 20 | 4.4% | |
| **Distance to Academic Hospital** | | | | | 0.0001 | | | | | 0.1477 |
| Less than 20 Miles | 1111 | 93.5% | 77 | 6.5% | | 1123 | 94.5% | 65 | 5.5% | |
| 20–50 Miles | 754 | 97.2% | 22 | 2.8% | | 745 | 96.1% | 31 | 4.0% | |
| 50–100 Miles | 1707 | 96.3% | 65 | 3.7% | | 1701 | 96.0% | 71 | 4.0% | |
| 100+ Miles | 1176 | 96.3% | 45 | 3.7% | | 1175 | 96.2% | 46 | 3.8% | |

**Table 3.** Factors associated with having EGFR somatic mutation testing in Stage IIIb–Stage IV NSCLC patients.

| Modeling Had EGFR Testing | | |
|---|---|---|
| **Variable** | **OR (95% CI)** | **P-Value** |
| **Age (ref = 75+)** | | 0.0001 |
| 20–49 | 4.15 (2.17–7.91) | |
| 50–64 | 1.76 (1.16–2.67) | |
| 65–74 | 1.39 (0.98–1.98) | |
| **Sex (ref = Male)** | | 0.0142 |
| Female | 1.44 (1.08–1.93) | |
| **Appalachian Status (ref = Non-Appalachia/Metro)** | | 0.0011 |
| Appalachian/Metro | 0.67 (0.28–1.59) | |
| Appalachian/Non-Metro | 0.51 (0.36–0.73) | |
| Non-Appalachian/Non-Metro | 0.60 (0.40–0.89) | |
| **Year of Diagnosis (ref = 2007)** | | <0.0001 |
| 2008 | 3.81 (0.43–34.68) | |
| 2009 | 22.30 (3.00–165.41) | |
| 2010 | 58.56 (8.12–422.26) | |
| 2011 | 113.47 (15.81–814.21) | |
| **Insurance (ref = Private)** | | <0.0001 |
| Medicaid | 0.19 (0.09–0.40) | |
| Medicare | 0.61 (0.44–0.84) | |
| **Smoking (ref = No)** | | 0.0266 |
| Yes | 0.54 (0.32–0.91) | |
| Unknown | 0.83 (0.42–1.66) | |

OR = odds ratio; CI = confidence interval; (ref) = reference variable

**Table 4. Factors associated with the receiving erlotinib in Stage IIIb- Stage IV NSCLC patients.**

| Modeling Receive Erlotinib | | |
|---|---|---|
| **Variable** | **OR (95% CI)** | **P-Value** |
| **Age (ref = 75+)** | | 0.0077 |
| 20–49 | 2.05 (1.02–4.14) | |
| 50–64 | 1.97 (1.31–2.95) | |
| 65–74 | 1.56 (1.10–2.21) | |
| **Sex (ref = Male)** | | 0.0045 |
| Female | 1.49 (1.13–1.97) | |
| **Insurance (ref = Private)** | | 0.0074 |
| Medicaid | 0.55 (0.33–0.93) | |
| Medicare | 0.63 (0.46–0.87) | |
| **Poverty (ref = Low)** | | 0.0081 |
| Moderate | 1.90 (1.24–2.91) | |
| High | 1.84 (1.22–2.79) | |
| Very High | 1.33 (0.85–2.09) | |

OR = odds ratio; CI = confidence interval; (ref) = reference variable

**Table 5. Cox-regression survival analysis for Stage IIIb-IV NSCLC patients.**

| Variable | HR (95% CI) | P-Value |
|---|---|---|
| **Age (ref = 75+)** | | <0.0001 |
| 20–49 | 0.65 (0.55–0.77) | |
| 50–64 | 0.76 (0.70–0.83) | |
| 65–74 | 0.79 (0.74–0.85) | |
| **Sex (ref = Male)** | | <0.0001 |
| Female | 0.88 (0.83–0.93) | |
| **Appalachian Status (ref = Non-Appalachia/Metro)** | | 0.0372 |
| Appalachian/Metro | 1.09 (0.93–1.28) | |
| Appalachian/Non-Metro | 1.10 (0.97–1.25) | |
| Non-Appalachian/Non-Metro | 1.13 (1.04–1.23) | |
| **Insurance (ref = Private)** | | 0.0053 |
| Medicaid | 1.17 (1.05–1.31) | |
| Medicare | 1.11 (1.03–1.19) | |
| **Poverty (ref = Low Poverty)** | | 0.0516 |
| Moderate | 1.10 (1.02–1.20) | |
| High | 0.98 (0.90–1.07) | |
| Very High | 1.01 (0.88–1.16) | |
| **Stage (ref = Stage IV)** | | 0.0320 |
| Stage IIIb and effusion | 0.88 (0.78–0.99) | |
| **Charlson Comorbidity Index (ref = 3+)** | | <0.0001 |
| 0 | 0.76 (0.70–0.83) | |
| 1 | 0.82 (0.75–0.90) | |
| 2 | 0.85 (0.77–0.95) | |
| **EGFR Test (ref = No Test)** | | 0.0003 |
| Received Test | 0.77 (0.67–0.89) | |
| **Erlotinib Drug (ref = No Drug)** | | <0.0001 |
| Received Drug | 0.62 (0.54–0.71) | |

HR = hazard ratio; CI = confidence interval; (ref) = reference variable

Finally, those patients receiving the EGFR test had a significantly increased likelihood of survival compared to those who had not received the test (HR = 0.77, CI = 0.67 to 0.89, P = 0.0030). Similarly, those patients that received erlotinib had an increased likelihood of survival compared to those who did not receive the drug (HR = 0.62, CI = 0.54 to 0.71, P = <0.0001).

Kaplan-Meier survival estimates indicate that those patients receiving EGFR testing had an increased survival probability compared to those that did not receive EGFR testing (Fig 1a). Those that received erlotinib also had an increased survival probability compared to those patients not receiving the drug, especially during the 0 to 20 month time period (Fig 1b).

## Discussion

The original publications outlining the sensitivity of EGFR-positive NSCLC tumors to tyrosine kinase inhibitors (TKI) were published in 2004, and the first EGFR assay was commercialized in 2005 [5–7]. Despite this, our analysis found that during the years 2007–2011, EGFR testing rates remained low. Erlotinib was approved as a second-line therapy in 2004 for metastatic NSCLC regardless of EGFR status, and its rate of use was also minimal in the years examined [8].

While EGFR testing rates have increased over time, still not all eligible patients receive testing. A study evaluating NSCLC patients seen in community medical oncology practices in New Jersey and Maryland showed between 2013 to 2015, 59% of eligible patients were tested for EGFR mutations, while a second study using data from a national, private health insurance company found testing rates to be around 61% between the years of 2010 to 2012 [19, 20]. In comparison, testing rates in Kentucky were substantially lower during this same time period, with 7% of eligible patients tested in 2010 and 12% tested in 2011, highlighting disparities between urban, privately insured individuals and rural, Medicare recipients. The time lag between the first publications in 2004 and the uptake of the EGFR test and erlotinib use could be due to a number of causes, both at a population level and due to individual patient characteristics. On a population level, the Centers for Medicare and Medicaid Services (CMS) did not approve reimbursement of the EGFR test until 2008, and ASCO and NCCN did not update their guidelines until 2010 [9, 10, 21]. Additionally, FDA-approved indications for erlotinib have changed multiple times since its approval in 2004, with 2013 being the first time it was indicated specifically for those patients with EGFR mutations. Finally, as each year passed, patients were more likely to receive the test compared to 2007, the first year of our analysis, suggesting wider implementation of testing over time.

Our analysis found patient level factors that further influenced testing rates and erlotinib prescribing. Younger patients and female patients were more likely to be tested for EGFR mutations and to receive erlotinib. This is possibly due to EGFR mutations occurring more frequently in younger NSCLC patients as well as in women [22, 23]. Factors that contributed to patients being less likely to receive the EGFR test were enrollment in Medicare or Medicaid and living in a rural area regardless of Appalachian status. Patients enrolled in Medicare or Medicaid and those living in high poverty areas were also significantly less likely to receive the drug.

While population factors, including delays in reimbursement, development of guidelines, and evolving FDA indications likely influenced uptake in Kentucky, we anticipate that patient characteristics associated with decreased testing are over-represented in our population and contribute to the lower than national average testing rates over the same time period. Specifically, our population contained a higher number of Medicare/Medicaid patients compared to the studies described above. In addition, Kentucky's poverty rate is significantly higher than

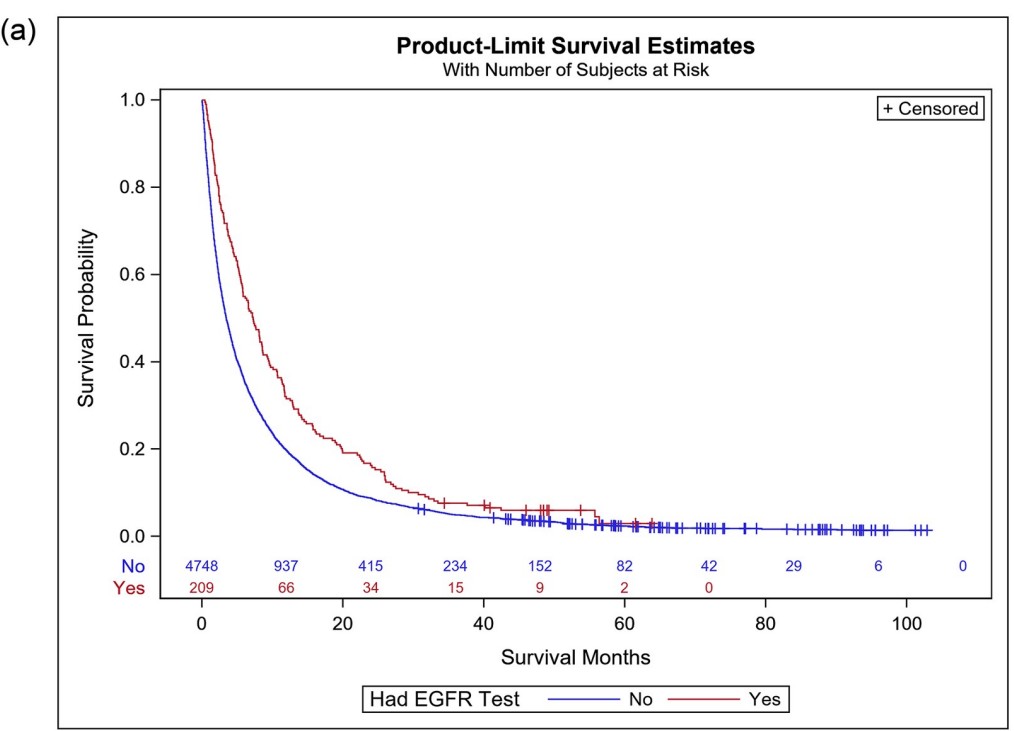

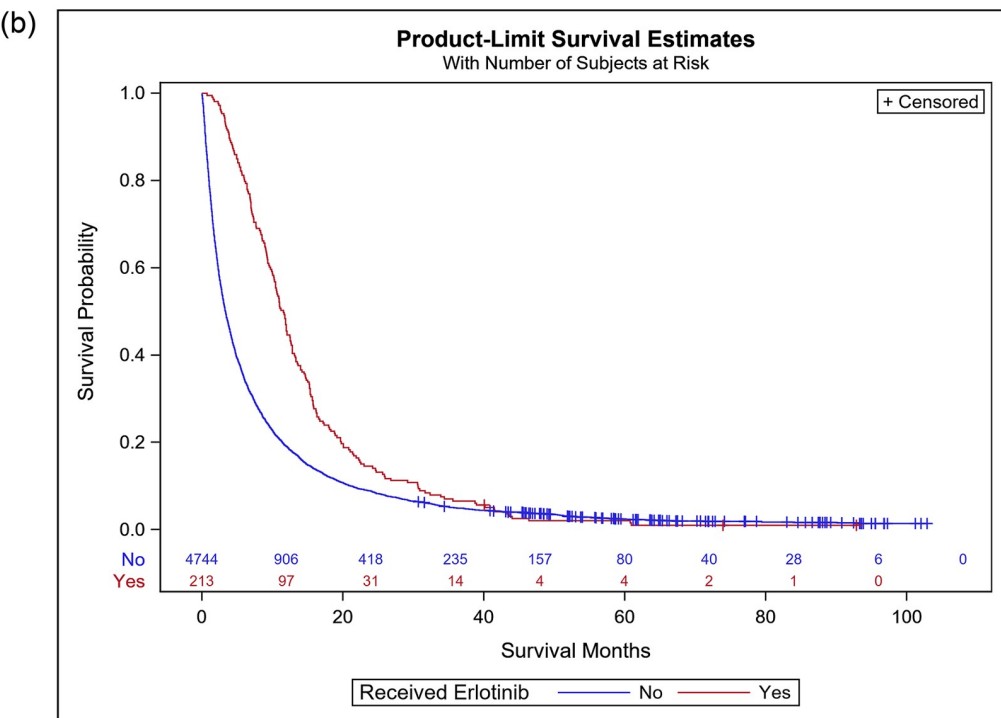

**Fig 1.** a. Kaplan-Meier survival curves for NSCLC patients by EGFR testing status b. Kaplan-Meier survival curves for NSCLC patients by erlotinib status.

the national average (KY = 18.3%, national = 14.6%), with several counties in Appalachia reaching 35–40% [24]. This could result in significant health disparities compared with national or less rural populations.

Patients that received EGFR testing had increased survival compared to those who did not. As expected, younger age, female gender, lower stage, and less comorbidities were associated with improved survival. Other factors associated with better survival included having private insurance and living in a non-Appalachia, metropolitan area. Since testing itself should not impact survival, this is likely due to those patients receiving better overall healthcare, related to better access to care or better insurance coverage. Nationally, patients with cancer living in rural areas have worse outcomes when compared to those living in urban areas, related to income and access inequalities, and highlighting these disparities in our population and suggesting better overall healthcare in these patients as a proxy for increasing their chances for survival.[25, 26] Those patients that received erlotinib also had a significantly better chance of survival. This could be an effect of the drug or that patients with EGFR mutant positive NSCLC have an overall better prognosis than those who do not [27].

To our knowledge, this is the largest description of the use of precision medicine in a predominantly rural population and the first to show the impact of precision medicine implementation on patient outcomes. It is also the first to look at precision medicine in Appalachia, a predominantly impoverished and disparate population. Importantly, we demonstrate the uptake of precision medicine in a rural population and suggest that new testing and treatment strategies would similarly lag behind urban and academic medical centers. While the management of NSCLC has changed over the intervening years, this analysis has several advantages, including mature survival data and a comprehensive assessment of implementation over an extended time-period. In addition, data was collected longitudinally using a registry-based cohort, which allows for a large sample size and minimizes selection bias. Lastly, at the time that the data was collected, erlotinib was the only EGFR inhibitor available and NGS panel testing was not performed in Kentucky, which provides the opportunity to observe the implementation of a single precision medicine test and treatment in a population without competing interventions.

This study is not without its limitations. The EGFR status or the prior treatment history of the tested individuals is unknown and so we cannot assess the appropriateness of erlotinib prescribing. Only EGFR testing within three months, and erlotinib prescribing within one year of diagnosis were assessed. It is possible that patients received the testing or the drug outside of this time window, but the median survival of late stage lung cancer at the time of data collection was only twelve months, and we anticipate few, if any patients were missed. In addition, while the number of cases of lung cancer diagnosed in Kentucky were drawn from a population-based cancer registry, the analysis of erlotinib prescribing and EGFR testing was conducted with a linked insurance claims database. Therefore, uninsured patients were not included in the analysis, which represents a selection bias against the poorest end of the spectrum. We anticipate that the 5% of uninsured Kentucky patients with lung cancer were even less likely to receive testing or erlotinib therapy and to have poorer outcomes [28]. Linked claims were only available for the time period reported, so while these results do not reflect current practice patterns, the study presented the opportunity to study the implementation of a single precision medicine intervention without competing interventions. We hypothesize that precision medicine interventions continue to lag in rural communities and this highlights the need for further study. Lastly, we could not measure physician-related factors such as available resources and education.

In conclusion, EGFR testing and prescribing of erlotinib occurred at a low rate in in Kentucky. While population factors likely contributed, patient level factors including residing in

rural areas and type of insurance were associated with decreased use and reduced survival, highlighting rural disparities in the implementation of cancer precision medicine.

## Author Contributions

**Conceptualization:** Susanne M. Arnold, Jill M. Kolesar.

**Formal analysis:** Bin Huang, Quan Chen.

**Funding acquisition:** Jill M. Kolesar.

**Project administration:** Susanne M. Arnold, Jill M. Kolesar.

**Writing – original draft:** Kara L. Larson, Marissa Schuh.

**Writing – review & editing:** Kara L. Larson, Bin Huang, Quan Chen, Thomas Tucker, Susanne M. Arnold, Jill M. Kolesar.

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
