## [Decision Letter · Decision Letter 0]

23 Apr 2020

PONE-D-20-04879

EGFR testing and erlotinib use in non-small cell lung cancer patients in Kentucky

PLOS ONE

Dear Dr Kolesar,

Thank you for submitting your manuscript to PLOS ONE. After careful consideration, we feel that it has merit but does not fully meet PLOS ONE’s publication criteria as it currently stands. Therefore, we invite you to submit a revised version of the manuscript that addresses the points raised during the review process.

We would appreciate receiving your revised manuscript by Jun 07 2020 11:59PM. To enhance the reproducibility of your results, we recommend that if applicable you deposit your laboratory protocols in protocols.io, where a protocol can be assigned its own identifier (DOI) such that it can be cited independently in the future. For instructions see: http://journals.plos.org/plosone/s/submission-guidelines#loc-laboratory-protocols

We look forward to receiving your revised manuscript.

Kind regards,

Randall J. Kimple

Academic Editor

PLOS ONE

2. In the ethics statement in the manuscript and in the online submission form, please provide additional information about the patient records used in your retrospective study. Specifically, please ensure that you have discussed whether all data were fully anonymized before you accessed them and/or whether the IRB or ethics committee waived the requirement for informed consent. If patients provided informed written consent to have data from their medical records used in research, please include this information.

3. Thank you for including your ethics statement:  "University of Kentucky IRB, Written consent, IRB #51483".   

Please amend your current ethics statement to confirm that your named institutional review board or ethics committee specifically approved this study.

Reviewers' comments:

Reviewer's Responses to Questions

**Comments to the Author**

1. Is the manuscript technically sound, and do the data support the conclusions?

Reviewer #1: Yes

Reviewer #2: Yes

2. Has the statistical analysis been performed appropriately and rigorously? 

Reviewer #1: Yes

Reviewer #2: Yes

3. Have the authors made all data underlying the findings in their manuscript fully available?

Reviewer #1: Yes

Reviewer #2: Yes

4. Is the manuscript presented in an intelligible fashion and written in standard English?

Reviewer #1: Yes

Reviewer #2: Yes

5. Review Comments to the Author

Reviewer #1: This paper examined the prevalence of EGFR testing and erlotinib use in Kentucky using registry and insurance claim data. The manuscript is clear and well-written with appropriate statistical design. This work may raise community awareness and lead to improved adoption of guidelines for the management of NSCLC; both are strengths.

Several concerns are included below regarding the design and applicability of this work:

1. The cohort examined in this study was from 2007-2011. Since then, management for NSCLC, especially testing and treatment for EGFR-driven disease, has undergone major transformations. With the publication of the FLAURA trial in 2018 and subsequent FDA approval of osimertinib for treatment-naïve EGFR mutated NSCLC, the standard of care for front-line therapy has changed and erlotinib is no longer the recommendated therapy for advanced EGFR-driven NSCLC. The relevance and applicability of this study are questionable given this evolving practice pattern. If the authors could update their cohort to reflect the change in practice pattern, it would significantly strengthen the work.

2. Along these lines, if the authors are able to include more recent data, i.e. post 2016, it would be informative and enhance the story to look at testing pattern and drug usage for other mutation-driven NSCLC such as ALK. Crizotinib was approved in 2016 so the authors would need to have access to registry and insurance claim data post-2016.

3. Similar studies looking at this question have been published previously with more recent data (after 2011), with similar conclusions. The authors even cited one such study in their references. Thus, it is hard to differentiate the novelty of the current work from its predecessors.

4. It would be more informative if the authors were able determine that among the patients who received EGFR testing, how many tested positive? Among those who tested positive, how many received erlotinib.

5. Confounders and biases, some the authors have addressed in the conclusion:

a. Study did not address why utilization of EGFR testing and erlotinib is so low. Was this due to physician education, patient understanding, availability of testing, lack of insurance coverage? The design of the study examined mostly patient-specific and possibly insurance factors but did not address availability of resources or physician-related factors.

b. All patients in this study had insurance coverage. So those who did not were excluded. Unclear how this reflects the broader population of Kentucky.

c. The % of EGFR testing and erlotinib prescription may be falsely low compared to other similar studies or even the national average because the design of the study captured all patients rather than patients who fit the demographics of EGFR-driven disease. Since the general NSCLC patient population in Kentucky probably has lower EGFR prevalence compared to some areas in the US, i.e. West Coast, if looking at all-comers, both testing and treatment may be lower because the prevalence of EGFR mutation is lower.

d. Table 5 Cox-regression survival analysis I do not think that EGFR testing and erlotinib being statistically significant in this model are meaningful because, as the authors pointed out, testing is a likely surrogate for receiving guideline-appropriate care, and EGFR-driven disease is relatively more indolent with better prognosis. Both of these have favorable impact on survival.

6. Minor: please state explicitly how many patients were actually included in the cohort.

Reviewer #2: This study analyzed factors associated with EGFR testing and erlotinib prescribing in Kentucky from 2007 to 2011. The analysis used the Kentucky Cancer Registry linked with health claims from Medicaid, Medicare and private insurance groups. The study concludes that EGFR testing and prescribing of erlotinib occurred at a low rate in Kentucky and factors including residing in rural areas and type of insurance were associated with decreased use and reduced survival. While the methodology appears, appropriate, my main concern is the overall relevance of this study in 2020.

This paper looks at EGFR testing and erlotinb use during a time when EGFR inhibitors were still under clinical investigation and not FDA approved as front line therapy. Practice patterns of oncologists were still adjusting as new data came out. Multiple previous papers, which the authors have cited, have already been published on this topic showing the slow rate of testing and the obstacles of implementing EGFR testing. The analysis has multiple limitations as outlined in the second from last paragraph. It is therefore hard to draw firm conclusions from the data. Not sure, how this data is relevant in 2020 or how it would be used to advanced patient care or current public health policy in Kentucky. The conclusion that rural areas and poverty are barriers to providing health care have already been well documented. An analysis of current data and/or an analysis that examines specific barriers to EGFR testing (such as state policy on testing or laboratory specific barriers or educational programs for oncologists) would have been more impactful.

Comments to be addressed:

• It is unclear what this this study contributes to the field. The analysis appears to be 10 years too late to impact public health policy on precision medicine. Can you explain why this analysis is relevant in 2020? What is the status of EGFR or broad genomic testing for NSCLC in Kentucky in 2020?

• Can the authors speak more about access to EGFR testing in Kentucky? Was testing being done in Kentucky or was it being sent out of state? How many centers in Kentucky were doing EGFR testing during this time? Was there a state effort during this time to assist in EGFR testing?

• Can the authors discuss the barriers that individual oncologists faced with EGFR testing? Was there slow dissemination of knowledge among the healthcare team? What is the distribution of oncologists in regards to rural and metro locations?

• It is unclear if you are analyzing front line use of erlotinib or second-line use or erlotinib. Please clarify in the methods.

• Why was gefitinib not included in the analysis? It was FDA approved in 2003 as second line therapy in metastatic NSCLC.

• Discussion paragraph #5: The survival difference seen in those that had EGFR testing can also be attributed to younger age and because a higher proportion were most likely EGFR mutated and actually derived benefit from erlotinib.

6. PLOS authors have the option to publish the peer review history of their article (what does this mean?). If published, this will include your full peer review and any attached files.

Reviewer #1: No

Reviewer #2: No

---

## [Author Response · Author response to Decision Letter 0]

10 Jun 2020

Author response to reviewer comments

 Author response: Revised as suggested

2. In the ethics statement in the manuscript and in the online submission form, please provide additional information about the patient records used in your retrospective study. Specifically, please ensure that you have discussed whether all data were fully anonymized before you accessed them and/or whether the IRB or ethics committee waived the requirement for informed consent. If patients provided informed written consent to have data from their medical records used in research, please include this information.

 Author response: revised as suggested

3. Thank you for including your ethics statement: "University of Kentucky IRB, Written consent, IRB #51483". 

Please amend your current ethics statement to confirm that your named institutional review board or ethics committee specifically approved this study.

 Author response: revised as suggested

Author Response. Raw data cannot be shared because they are both potentially identifying and contain sensitive patient data, including geographic location, dates of diagnosis and dates of testing and receiving a medication. In addition, there are contractual agreements between the University of Kentucky and the Kentucky Cancer Registry precluding data sharing. Any requests for data must be submitted to:

Jacyln K. McDowell, Epidemiologist, Kentucky Cancer Registry

2365 Harrodsburg Rd, Suite A230

Lexington, KY 40504

859-218-2228

 Review Comments to the Author

Reviewer #1: This paper examined the prevalence of EGFR testing and erlotinib use in Kentucky using registry and insurance claim data. The manuscript is clear and well-written with appropriate statistical design. This work may raise community awareness and lead to improved adoption of guidelines for the management of NSCLC; both are strengths.

Several concerns are included below regarding the design and applicability of this work:

1. The cohort examined in this study was from 2007-2011. Since then, management for NSCLC, especially testing and treatment for EGFR-driven disease, has undergone major transformations. With the publication of the FLAURA trial in 2018 and subsequent FDA approval of osimertinib for treatment-naïve EGFR mutated NSCLC, the standard of care for front-line therapy has changed and erlotinib is no longer the recommendated therapy for advanced EGFR-driven NSCLC. The relevance and applicability of this study are questionable given this evolving practice pattern. If the authors could update their cohort to reflect the change in practice pattern, it would significantly strengthen the work. 

Author response: The discussion was revised as followd: “To our knowledge, this is the largest description of the use of precision medicine in a predominantly rural population and the first to show the impact of precision medicine implementation on patient outcomes. It is also the first to look at precision medicine in Appalachia, a predominantly impoverished and disparate population. Importantly, we demonstrate the uptake of precision medicine in a rural population and suggest that new testing and treatment strategies would similarily lag behind urban and academic medical centers. While the management of NSCLC has changed over the intervening years, this analysis has several advantages, including mature survival data and a comprehensive assessment of implementation over an extended time-period. In addition, data was collected longitudinally using a registry-based cohort, which allows for a large sample size and minimizes selection bias. Lastly, at the time that the data was collected, erlotinib was the only EGFR inhibitor available and NGS panel testing was not performed in Kentucky, which provides the opportunity to observe the implementation of a single precision medicine test and treatment in a population without competing interventions.

2. Along these lines, if the authors are able to include more recent data, i.e. post 2016, it would be informative and enhance the story to look at testing pattern and drug usage for other mutation-driven NSCLC such as ALK. Crizotinib was approved in 2016 so the authors would need to have access to registry and insurance claim data post-2016. 

Author response: Please see response to comment 1, above. 

3. Similar studies looking at this question have been published previously with more recent data (after 2011), with similar conclusions. The authors even cited one such study in their references. Thus, it is hard to differentiate the novelty of the current work from its predecessors. 

Author response: We suggest that the novelty of the finding is that testing was lower in a rural region with known health disparities over the same time period and have revised the discussion as follows (bolded=new):

In comparison, we found that the testing rates in Kentucky were substantially lower during this same time period, with 7% of eligible patients tested in 2010 and 12% tested in 2011, highlighting disparities between urban, privately insured individuals and rural, Medicare recipients.

4. It would be more informative if the authors were able determine that among the patients who received EGFR testing, how many tested positive? Among those who tested positive, how many received erlotinib.

Author response: We agree, and since this data is not available we included it as a limitation

5. Confounders and biases, some the authors have addressed in the conclusion:

a. Study did not address why utilization of EGFR testing and erlotinib is so low. Was this due to physician education, patient understanding, availability of testing, lack of insurance coverage? The design of the study examined mostly patient-specific and possibly insurance factors but did not address availability of resources or physician-related factors.

Author response: We agree, and since this data is not available we revised the limitation to include the following. 

Lastly, we could not measure physician-related factors such as available resources and education.

b. All patients in this study had insurance coverage. So those who did not were excluded. Unclear how this reflects the broader population of Kentucky. 

Author response: The methods were revised as follows: Over this time period, 5.3% of diagnosed cases occurred in uninsured individuals who were excluded from the analysis. 

c. The % of EGFR testing and erlotinib prescription may be falsely low compared to other similar studies or even the national average because the design of the study captured all patients rather than patients who fit the demographics of EGFR-driven disease. Since the general NSCLC patient population in Kentucky probably has lower EGFR prevalence compared to some areas in the US, i.e. West Coast, if looking at all-comers, both testing and treatment may be lower because the prevalence of EGFR mutation is lower.

Author response: Thank-you for your comment. While we know today that EGFR mutation frequency is lower in Kentucky than in other populations, this was not known during the study time period and national guidelines has not recommended testing sub-sets for EGFR mutations and the referenced studies we are comparing to did not exclude patients based on clinical characteristics.

d. Table 5 Cox-regression survival analysis I do not think that EGFR testing and erlotinib being statistically significant in this model are meaningful because, as the authors pointed out, testing is a likely surrogate for receiving guideline-appropriate care, and EGFR-driven disease is relatively more indolent with better prognosis. Both of these have favorable impact on survival. 

Author response: We respectfully disagree. Disparities in survival between rural and urban cancer patients is well known and likely related to access and income inequalities. Lack of EGFR testing (and other standard of care treatments) in our population is likely contributing to the survival disadvantage.

The following was added to the discussion: Nationally, patients with cancer living in rural areas have worse outcomes when compared to those living in urban areas, related to income and access inequalities, and highlighting these disparities in our population.

The following references were cited: 

Henley SJ, Anderson RN, Thomas CC, Massetti GM, Peaker B, Richardson LC. Invasive cancer incidence, 2004-2013, and deaths, 2006-2015, in nonmetropolitan and metropolitan counties—United States. MMWR Surveill Summ. 2017;66(14):-. doi:10.15585/mmwr.ss6614a1 

Iglehart JK. The challenging quest to improve rural health care. N Engl J Med. 2018;378(5):473-479. doi:10.1056/NEJMhpr1707176

6. Minor: please state explicitly how many patients were actually included in the cohort. 

Author response: The methods were revised as follows: The final cohort included 4957 individuals.

Reviewer #2: This study analyzed factors associated with EGFR testing and erlotinib prescribing in Kentucky from 2007 to 2011. The analysis used the Kentucky Cancer Registry linked with health claims from Medicaid, Medicare and private insurance groups. The study concludes that EGFR testing and prescribing of erlotinib occurred at a low rate in Kentucky and factors including residing in rural areas and type of insurance were associated with decreased use and reduced survival. While the methodology appears, appropriate, my main concern is the overall relevance of this study in 2020.

This paper looks at EGFR testing and erlotinb use during a time when EGFR inhibitors were still under clinical investigation and not FDA approved as front line therapy. Practice patterns of oncologists were still adjusting as new data came out. Multiple previous papers, which the authors have cited, have already been published on this topic showing the slow rate of testing and the obstacles of implementing EGFR testing. The analysis has multiple limitations as outlined in the second from last paragraph. It is therefore hard to draw firm conclusions from the data. Not sure, how this data is relevant in 2020 or how it would be used to advanced patient care or current public health policy in Kentucky. The conclusion that rural areas and poverty are barriers to providing health care have already been well documented. An analysis of current data and/or an analysis that examines specific barriers to EGFR testing (such as state policy on testing or laboratory specific barriers or educational programs for oncologists) would have been more impactful.

Comments to be addressed:

• It is unclear what this this study contributes to the field. The analysis appears to be 10 years too late to impact public health policy on precision medicine. Can you explain why this analysis is relevant in 2020? What is the status of EGFR or broad genomic testing for NSCLC in Kentucky in 2020? 

Author response: Please see response to comment 1, reviewer 1, above. 

• Can the authors speak more about access to EGFR testing in Kentucky? Was testing being done in Kentucky or was it being sent out of state? How many centers in Kentucky were doing EGFR testing during this time? Was there a state effort during this time to assist in EGFR testing? 

Author response: Thank-you for your comment, but physician related factors this outside the scope of the manuscript. We have revised the discussion to include this as a limitation. 

• Can the authors discuss the barriers that individual oncologists faced with EGFR testing? Was there slow dissemination of knowledge among the healthcare team? What is the distribution of oncologists in regards to rural and metro locations? 

Author response: Thank-you for your comment, but physician related factors this outside the scope of the manuscript. We have revised the discussion to include this as a limitation. 

• It is unclear if you are analyzing front line use of erlotinib or second-line use or erlotinib. Please clarify in the methods. 

Author response: The methods were revised as follows (new text bold): Drug claims were captured within one year of diagnosis and could have been any line treatment.

• Why was gefitinib not included in the analysis? It was FDA approved in 2003 as second line therapy in metastatic NSCLC. 

Author response: We did assess gefitinib and crizotinib use in the claims cohort, but since no gefitinib was prescribed and less than 5 crizotinib claims were identified we did not include them in the analysis.

• Discussion paragraph #5: The survival difference seen in those that had EGFR testing can also be attributed to younger age and because a higher proportion were most likely EGFR mutated and actually derived benefit from erlotinib. 

Author response: We respectfully disagree, the survival analysis (Cox-regression model) controls for other factors associated with survival and having EGFR testing in the model is an independent predictor of survival.

---

## [Decision Letter · Decision Letter 1]

23 Jun 2020

PONE-D-20-04879R1

EGFR testing and erlotinib use in non-small cell lung cancer patients in Kentucky

PLOS ONE

Dear Dr. Kolesar,

Thank you for submitting your manuscript to PLOS ONE. After careful consideration, we feel that it has merit but does not fully meet PLOS ONE’s publication criteria as it currently stands. Therefore, we invite you to submit a revised version of the manuscript that addresses the points raised during the review process.

We look forward to receiving your revised manuscript.

Kind regards,

Randall J. Kimple

Academic Editor

PLOS ONE

Reviewers' comments:

Reviewer's Responses to Questions

**Comments to the Author**

1. If the authors have adequately addressed your comments raised in a previous round of review and you feel that this manuscript is now acceptable for publication, you may indicate that here to bypass the “Comments to the Author” section, enter your conflict of interest statement in the “Confidential to Editor” section, and submit your "Accept" recommendation.

Reviewer #1: (No Response)

Reviewer #2: (No Response)

2. Is the manuscript technically sound, and do the data support the conclusions?

Reviewer #1: Yes

Reviewer #2: Partly

3. Has the statistical analysis been performed appropriately and rigorously? 

Reviewer #1: Yes

Reviewer #2: Yes

4. Have the authors made all data underlying the findings in their manuscript fully available?

Reviewer #1: Yes

Reviewer #2: Yes

5. Is the manuscript presented in an intelligible fashion and written in standard English?

Reviewer #1: Yes

Reviewer #2: Yes

6. Review Comments to the Author

Reviewer #1: Since both reviewers raised the concern of novelty and relevance of the study, please ask the authors to explain why they did not choose to expand the analysis using the Kentucky Cancer Registry to years beyond 2011 to reflect the changing practice patterns in EGFR-driven NSCLC, rather than simply rephrasing their discussion. If this is feasible, it should done.

Reviewer #2: Table 4: Since EGFR screening is associated with younger non-smoking women (most likely a surrogate for selecting appropriate patients for testing), is it possible that younger non-smoking women were more likely to have private insurance and lower poverty? Have you performed any statistical tests to see if these are associated? If so, it is possible that being a young non-smoker woman is the most important variable associated with testing and not necessary access to testing.

Discussion paragraph 5: The conclusion that that the improved survival with EGFR testing and erlotinib "is likely due to those patients receiving better overall healthcare, relating to care access, insurance coverage or poverty status" is flawed and not supported by your data. Overall survival is most likely better because the patients that received EGFR testing were more likely to have an EGFR mutation and thus obtained benefit from erlotinib. You state that this is not the case because only 10-15% would be positive for the mutation. This 10-15% is for the general public. It is most likely that those receiving EGFR testing in your population have a high chance of having an EGFR mutation given they they are predominately younger non-smoking women. Also, a population enriched with 10-15% of patients with EGFR mutations receiving erlotinib can still be responsible for the overall survival benefit. If you want to make a conclusion that access to care leads to better survival that is fine. But use the metropolitan vs rural and medicaid/medicare vs private insurance analysis instead of the EGFR/erlotinib analysis to support this conclusion.

7. PLOS authors have the option to publish the peer review history of their article (what does this mean?). If published, this will include your full peer review and any attached files.

Reviewer #1: No

Reviewer #2: No

---

## [Author Response · Author response to Decision Letter 1]

17 Jul 2020

Reviewer #1: Since both reviewers raised the concern of novelty and relevance of the study, please ask the authors to explain why they did not choose to expand the analysis using the Kentucky Cancer Registry to years beyond 2011 to reflect the changing practice patterns in EGFR-driven NSCLC, rather than simply rephrasing their discussion. If this is feasible, it should done.

Author response: The linked claims database was created with a funded research collaboration that covered the time period reported, therefore it is not feasible to update the analysis. 

The following was added to the limitations paragraph: “Linked claims were only available for the time period reported, so while these results do not reflect current practice patterns, the study presented the opportunity to study the implementation of a single precision medicine intervention without competing interventions. We hypothesize that precision medicine interventions continue to lag in rural communities and this highlights the need for further study.” 

Reviewer #2: Table 4: Since EGFR screening is associated with younger non-smoking women (most likely a surrogate for selecting appropriate patients for testing), is it possible that younger non-smoking women were more likely to have private insurance and lower poverty? Have you performed any statistical tests to see if these are associated? If so, it is possible that being a young non-smoker woman is the most important variable associated with testing and not necessary access to testing.

Author response: Table 4 shows the primary effect of significant factors associated with receiving erlotinib in a logistic regression analysis. The model identified independent association of each factor for the outcome variable. Hence the effects of insurance and poverty are significant regardless the status of age, gender or smoking. No changes were made to the manuscript. 

Discussion paragraph 5: The conclusion that that the improved survival with EGFR testing and erlotinib "is likely due to those patients receiving better overall healthcare, relating to care access, insurance coverage or poverty status" is flawed and not supported by your data. Overall survival is most likely better because the patients that received EGFR testing were more likely to have an EGFR mutation and thus obtained benefit from erlotinib. You state that this is not the case because only 10-15% would be positive for the mutation. This 10-15% is for the general public. It is most likely that those receiving EGFR testing in your population have a high chance of having an EGFR mutation given they they are predominately younger non-smoking women. Also, a population enriched with 10-15% of patients with EGFR mutations receiving erlotinib can still be responsible for the overall survival benefit. If you want to make a conclusion that access to care leads to better survival that is fine. But use the metropolitan vs rural and medicaid/medicare vs private insurance analysis instead of the EGFR/erlotinib analysis to support this conclusion.

Author response: The 5th paragraph of the discussion was revised as follows: 

As expected, younger age, female gender, lower stage, and less comorbidities were associated with improved survival. Other factors associated with better survival included having private insurance and living in a non-Appalachia, metropolitan area. Since testing itself should not impact survival, this is likely due to those patients receiving better overall healthcare, related to better care access to care or better insurance coverage.

---

## [Decision Letter · Decision Letter 2]

4 Aug 2020

EGFR testing and erlotinib use in non-small cell lung cancer patients in Kentucky

PONE-D-20-04879R2

Dear Dr. Kolesar,

We’re pleased to inform you that your manuscript has been judged scientifically suitable for publication and will be formally accepted for publication once it meets all outstanding technical requirements.

Kind regards,

Randall J. Kimple

Academic Editor

PLOS ONE

Additional Editor Comments (optional):

Reviewers' comments:

Reviewer's Responses to Questions

**Comments to the Author**

1. If the authors have adequately addressed your comments raised in a previous round of review and you feel that this manuscript is now acceptable for publication, you may indicate that here to bypass the “Comments to the Author” section, enter your conflict of interest statement in the “Confidential to Editor” section, and submit your "Accept" recommendation.

Reviewer #1: All comments have been addressed

Reviewer #2: All comments have been addressed

2. Is the manuscript technically sound, and do the data support the conclusions?

Reviewer #1: Yes

Reviewer #2: Yes

3. Has the statistical analysis been performed appropriately and rigorously? 

Reviewer #1: Yes

Reviewer #2: Yes

4. Have the authors made all data underlying the findings in their manuscript fully available?

Reviewer #1: Yes

Reviewer #2: Yes

5. Is the manuscript presented in an intelligible fashion and written in standard English?

Reviewer #1: Yes

Reviewer #2: Yes

6. Review Comments to the Author

Reviewer #1: While having more recent Kentucky Cancer Registry data would be better, all concerns were addressed by the authors.

Reviewer #2: The authors have addressed all of my concerns and have appropriately revised the manuscript. There are no new concerns.

7. PLOS authors have the option to publish the peer review history of their article (what does this mean?). If published, this will include your full peer review and any attached files.

Reviewer #1: No

Reviewer #2: No

---

## [Editor Report · Acceptance letter]

7 Aug 2020

PONE-D-20-04879R2 

EGFR testing and erlotinib use in non-small cell lung cancer patients in Kentucky 

Dear Dr. Kolesar:

I'm pleased to inform you that your manuscript has been deemed suitable for publication in PLOS ONE. Congratulations! Your manuscript is now with our production department. 

Kind regards, 

on behalf of

Dr. Randall J. Kimple 

Academic Editor

PLOS ONE